# Study on Electrical Explosion Properties of Cu/Ni Multilayer Exploding Foil Prepared by Magnetron Sputtering and Electroplating

**DOI:** 10.3390/mi11050528

**Published:** 2020-05-22

**Authors:** Fan Lei, Qin Ye, Shuang Yang, Qiubo Fu

**Affiliations:** 1Institute of Chemical Materials, Chinese Academy of Engineering Physics, Mianyang 621999, China; lf20200113@126.com (F.L.); baicaishuangr@163.com (S.Y.); 2College of Chemistry and Chemical Engineering, Southwest Petroleum University, Chengdu 610500, China; 201822000176@stu.swpu.edu.cn

**Keywords:** energy conversion efficiency, electric explosion, magnetron sputtering, Cu/Ni multilayer film, electroplating

## Abstract

The purpose of this study was to investigate the effects of the microstructure and properties of Cu/Ni multilayer films prepared by magnetron sputtering and electroplating on the electrical explosion performance of the films. In this study, Cu/Ni multilayer films of the same thickness were prepared by electroplating (EP) and magnetron sputtering (MS), and their morphology and crystal structure were characterized by scanning electron microscope (SEM) and transmission electron microscope (TEM). XRD was used to observe the crystal structure and size of the samples. In addition, the Cu/Ni multilayer film was etched into the shape of a bridge, and the electric explosion phenomenon in the same discharge circuit of the multilayer foil obtained by the two preparation processes was tested by an electric explosion performance test system. The resistance–time curve and the energy–resistance curve during the electric explosion process were analyzed and calculated. The results showed that compared with the multilayer film prepared by the MS method, the crystal size of the multilayer film prepared by the EP method is smaller and the interface of Cu/Ni is clearer. In the electric explosion experiment, the MS samples had earlier burst times, larger peak resistances, smaller peak energies and higher ionization voltages. Through observation of the morphology of the samples after the electric explosion and combination with gas ionization theory, the internal influencing factors of the peak voltage and the relative resistance of the two samples were analyzed. The influence of the multilayer film mixing layer thickness on the sample energy conversion efficiency was analyzed by modeling the microstructure of the multilayer film exploding foil and electric heating. The results show that the thicker the mixing layer is, the more energy is distributed on the Ni, the faster the resistance increases, and the higher the energy conversion efficiency.

## 1. Introduction

An exploding foil initiator (EFIS) is a short-pulse electric signal ignition device, and its core component is an explosive metal foil. Under the action of a high pulse current, the explosive metal foil completes the transformation of solid, liquid, gas and plasma in a very short time, forming a high-temperature and high-pressure plasma, thus promoting a flyer to impact and detonate the explosive [1,2,3,4,5]. Due to the high safety of impact plate detonators, since their introduction in a report from Lawrence Livermore National Laboratory in the United States, exploding metal foil initiators have rapidly become a popular research topic [6]. Traditional exploding foils generally use pure metal films such as Cu, Ag, and Au. Researchers have used electroplating, electron beam evaporation, magnetron sputtering and other methods to prepare pure metal foils and have conducted extensive and in-depth research in the performance of the corresponding electric explosive driven flyers [7,8,9,10,11,12,13].

For decades, methods for improving the performance of pure metal exploding foil have focused on its shape, thickness and the inductance of electrical circuits [14,15,16,17,18,19,20,21,22,23]. However, the electric explosion of pure metal foil has limitations: low energy conversion efficiency leads to high initiation energy, which is not conducive to miniaturization or low energy detonators. In recent years, for further improvement of the performance of flyers driven by foil electric explosions, multilayer materials stand out as candidates. A Cu/Al/CuO multilayer foil was prepared by Zhou [24] as well as work from Nicollet et al. [25] and Glavier et al. [26]. However, because the time scale of the chemical reaction between the materials was much longer than that of the electric explosion, the speed of the flyer was not improved. Morris et al. [27,28,29] carried out experimental research on the driving ability of flyers by Al/Ni multilayer foils and Cu foils. The results show that the additional energy obtained by the flyer is equivalent to the reaction energy of the multilayer film. Huang et al. [30] tested a Cu/Al/Ni multilayer foil and found that the addition of Al/Ni increased the feather size of the Cu plasma. 

Recently, researchers found that there was a premixed layer in multi-layer bridge foil. The properties of the premixed layer have a profound impact on the energy storage and release of the bridge foil. Wang [31] et al. carried out experimental research on the electric explosion performance and flyer driving performance of Al/Ni multilayer foil and analyzed the internal mechanism of the performance of Al/Ni multilayer foil, which was superior to that of a traditional Cu film exploding foil because of the crystallization reaction of Al and Ni. Their results also showed that the higher the premixed layer ratio was, the more energy was released from the Al/Ni alloy, and the faster the plasma flame advance. Wang et al. [32] studied the preparation of Al/Ni multilayer film exploding foils with different thicknesses by magnetron sputtering and characterized the heat release and reaction rate of combustion. The results show that the mutual diffusion rate of reactants has an important influence on the electric explosion performance of the film, and this rate is mainly affected by the thickness of the premixed layers.

We think that in addition to the changes of electrical explosion properties caused by alloying reaction, the electrical resistance of premixed layer is also the key factor to change the electrical explosion properties of multilayer foil. The resistivity and resistance growth rate of the foil determine the efficiency of its electro-thermal conversion, which affects the acceleration curve of flyer and the reliability of the EFIS on explosive ignition. Therefore, it is helpful to understand the mechanism of electric explosion of multilayered bridge foil by studying the relationship between the electric explosion performance and its microstructure. Additionally, this research is helpful for choosing suitable preparation methods of multilayer foil according to the demand.

In this paper, Cu/Ni multilayered foils prepared by electroplating and magnetron sputtering are used to study the influence of the resistance characteristics of the premixed layer with different thickness on the electrical explosion performance of multilayered foils. The microstructure of the multilayer foils is characterized. Combined with the analysis of the experimental results of the electrical explosion of multilayer foils, the electrical explosion performance of the multilayer foils prepared by the two processes is studied. Through theoretical modeling, the differences in the electrical explosion performance of multilayer foils with different thicknesses of premixed layer are analyzed, which lays a theoretical foundation for regulating the electrical explosion performance of multilayer foils through microstructure design.

## 2. Materials and Methods 

The two types of Cu/Ni foils used in this experiment were (a) samples prepared by the electroplating method (EP samples) and (b) samples prepared by the magnetron sputtering method (MS samples). As shown on Figure 1, the film thickness was 4 μm, and there were eight modulation layers in the Cu/Ni multilayer film. Each layer contained a Cu layer (200 μm) and a Ni layer (300 μm). For EP samples, Ni is plated on the glass substrate before the Cu is plated, so the top layer of the sample is a brown Cu layer. For MS samples, Cu is first plated on the ceramic substrate and then Ni is plated, so the top layer of the sample is silver Ni layer. In this way, the sample causes the material and the substrate to bond more firmly. The section morphology of the Cu/Ni multilayer film was analyzed by scanning electron microscope (SEM) (EVO18, ZEISS, Oberkochen, Germany), and the crystal microstructure of the multilayer film was observed by transmission electron microscope (Titan G2 60-300 (AC-TEM), FEI, Hillsboro, OR, USA). X-ray diffractometer (XRD, D8 advance, Bruker, Karlsruhe, Germany) was used to analyze the crystal structure and average cell size of multilayered foil.

To study the electrical explosion performance of the Cu/Ni multilayer film, the prepared Cu/Ni multilayer film was first photolithographed into a foil with a bridge area of 0.4 × 0.4 mm^2^. The energy deposition capacity in the process of Cu/Ni multilayer film explosion was studied by an electric explosion performance experimental system (as shown in Figure 2). A high-voltage pulse current was applied to both ends of the foil, and a high-voltage probe and oscilloscope were used to collect and record the electrical signals passing through the foil during charging. The sampling frequency of current and voltage was 1.25 GHz (the time interval is 0.8 ns), so as to ensure that the whole process of electric heating could be recorded accurately. The initial inductance of the loop was 121 nH when the EP sample was tested, and 118 nH when the MS sample was tested. The capacitor had a capacitance of 0.31 μF, a discharge cycle of 1.21 μs, and charging voltages of 2500 V, 3000 V, 3500 V, and 4000 V.

## 3. Results and Discussion

The microscopic images of the two sample sections obtained by SEM photography are shown in Figure 3. It can be seen that the Cu and Ni stratification of the EP sample is clearer, the limited transition layer can be observed, and the microstructure is more compact. Figure 4 shows TEM images of the microstructure of the two samples. It can be seen from the figure that the grains of the EP sample are very small, and the layered structure of the Cu layer and the Ni layer is obvious. Moreover, the thickness of the mixing layer is very small, and the film can be considered to have a multilayer structure. However, the grain size of the MS sample is relatively large, and the mixing layer is large, which makes it difficult to distinguish the layered structure in this sample. This sample can be considered to have a premixed structure of Cu grains and Ni grains.

The electroplating method used includes an electric field under the action of a metal salt solution; metal hydrate ions migrate to the electrode, the inner layer of the Helmholtz double layer accepts electrons, reducing in the electrode surface to form movable metal atoms, and with the organic glass substrate adsorption, the adsorbed atoms on the surface form a single-layer film that further proliferates. The magnetron sputtering method used is controlled by magnetic field excitation sputtering, causing metal atoms to fly to the substrate. In the process of the flight of the metal atoms, due to the presence of an inert gas, the metal atoms and inert gas collide at high frequency, decreasing the speed of the metal atoms, which ultimately form a layer of metal on the surface of an alumina ceramic substrate. The method of forming multilayer Cu/Ni films is based on the formation of a monolayer film, and the thickness of each layer of film is regulated by controlling the forming time of the monolayer metal film. Cu and Ni layers are alternately formed on the sample surface. For the electroplating, the metal atoms are electrodeposited on the substrate in solution, and for the magnetron sputtering, the metal atoms are in thin inert gas. Thus, for magnetic sputtering, the migration velocity of the metal atoms on the surface of the substrate is higher, and the diffusion ability of the atoms is stronger, making the pure metal film surface roughness higher and the mixing of the Cu and Ni layer interface worse.

The total thickness of EP sample is 4.02 ± 0.11 μm, and the average thickness of one composite layer is about 502.5 nm. The total thickness of MS sample is 3.88 ± 0.07 μm, and the average thickness of one composite layer is about 485.8 nm. For EP samples, we can clearly see the Ni layer from the TEM diagram, and measure its thickness as 220 ± 12 nm. The thickness between the two layers of Ni is 443 ± 7 nm. Assuming that the premixed layer is mixed according to the ratio of Ni:Cu = 3:2, the thickness of the premixed layer can be calculated to be about 37.3 nm. For EP samples, the proportion of premixed layer is about 2 × 37.3/442.6 = 17%. For MS samples, it is difficult to observe obvious stratification, so it is impossible to measure the premixed layer thickness by TEM.

To obtain the crystal state and grain size of the samples, the crystal structure and substrate of the multilayer films were analyzed by XRD. After removing the characteristic peak of ceramic and glass substrates, the results are as shown in Figure 5. Both Cu and Ni in the multilayer films prepared by the two methods are crystalline. The peaks of 2θ at 43.23°, 50.43° and 74.13° correspond to the diffraction peaks of Cu (111), Cu (200) and Cu (220), and the peaks at 44.59°, 51.95° and 76.60° correspond to the diffraction peaks of Ni (111), Ni (200) and Ni (220), respectively. The preferred orientations of Cu (111) and Ni (111) in MS samples are higher than those in EP samples. This is because the stronger the diffusion ability of metal atoms to the substrate surface, the better the growth of Cu film crystal. To obtain the grain size of the sample, we measured the full wave at half maximum (FWHM) of the strongest peak. The grain sizes of Cu and Ni are calculated using Equation (1) as shown in Table 1.
(1)D=0.89a/βcosθ
where *D* is the average diameter of the grains, a (0.154 nm) is the wavelength of the X-ray, β is the the FWHM of the strongest peak measured in radians. The grain size of MS sample is almost twice that of EP sample, which is consistent with TEM observation. It can be seen that the premixed layer thickness of EP sample is about the sum of the surface grains of Cu layer and Ni layer, so it can be inferred that the proportion of premixed layer in MS sample is about 34%.

By analyzing and processing the current and voltage curves obtained in the experiment, we obtained the evolution curve of the foil resistance with time, as shown in Figure 6. Comparing the resistance growth curves of the magnetron sputtering and electroplating methods shows that the resistance peaks of the magnetron samples appear earlier and are larger. The time difference between the two samples at the same voltage is ∆*t* = 17.0 ns (2500 V) and ∆*t* = 13.8 ns (4000 V). According to the analysis above, the difference between the inductances of the two samples is not significant at the same initial voltage, so the initial current growth rate is basically the same. The growth rate of each resistance curve shows that there is a rapid change point in the growth rate at several times the initial resistance, approximately *R*_1_ = 0.072 Ω. There is a sudden change in the resistivity change rate of solid and liquid during liquefaction at *R*_1_. It can be concluded that the time difference between the initial stage of sample electric heating (solid heating and melting stage of sample bridge area) is ∆*t* = 14.0 ns (2500 V) and ∆*t* = 13.2 ns (4000 V), which is very close to the time difference between the occurrence of the resistance peak. The main difference between the electric heating processes of the two samples occurs in the solid heating and melting stages. This is due to the difference between the microstructures of the two samples, which leads to the higher energy conversion efficiency of the magnetron sputtered samples during the electric heating process and the faster heating and melting of the solid. Thus, the explosion time is much earlier than that of the samples prepared by the electroplating method. In the liquid phase and gas phase of sample heating, the microstructure order of the sample is destroyed, and the structure difference of the sample disappears, which has little effect on the appearance time of the resistance peak of the sample.

According to Table 2, regardless of the initial voltage, the peak voltage of the MS sample is greater than that of the EP sample. Before the peak voltage appears, the gas produced by the bridge gasification is heated and expanded outward, and the electrons are localized to form a capacitor-like structure. The electric field strength between the non-vaporized electrodes increases until the following condition is met:(2)Ueδ/d≥W
where *U* is the voltage applied to the bridge, *d* is the minimum distance between the non-vaporized electrodes (shown on Figure 7), *δ* is the average free path of the electrons, *W* is the first ionization energy of the metal atom, and e is the charge of an electron. Due to the fixed type of atom, W is a constant. The possible reasons for the results in Table 1 are as follows: (1) the average molecular free path of the EP sample gas is larger than that of MS sample; (2) the vaporized area of the EP sample is smaller than that of MS sample, and the electrode distance d is smaller during ionization.

The above pictures on Figure 7 are microscope photos of the MS sample and EP sample after the electric explosion experiment. The current is horizontal, and the minimum distance between the non-vaporized electrodes is defined as d. It can be seen that the energy conversion rate of the MS sample is higher in the solid–liquid heating stage, so under the same voltage, compared with the EP sample, the region of the foil involved in melting is larger, and the energy is more dispersed. Thus, the area of complete gasification of the foil is smaller, and the distance d of its non-vaporized electrode is smaller. According to the above analysis, it can be seen that the reason the peak voltage of the MS sample is larger is not due to the second reason mentioned above. The results of the peak voltage comparison must be due to the average electron free path of the EP sample being larger than that of the MS sample. For a neutral high-temperature monatomic gas, the molecular free path is inversely proportional to the gas pressure and directly proportional to the temperature, meeting the following relationship:(3)$$\delta =\frac{kT}{p\sigma }$$
where k is the Boltzmann constant, *T* is the gas temperature, *p* is the gas pressure, and *σ* is the atomic collision cross-section. For the process of electric explosion, the peak temperature is less than 2 eV, far lower than the ionization energies of Cu and Ni. After the foil reaches the gasification temperature, the volume expands rapidly, the pressure decreases and the temperature increases. Therefore, the average free path of electrons increases with time, and the minimum field strength required for gas ionization decreases. Additionally, gas expansion leads to a decrease in conductivity and the formation of capacitor-like structures on the gas electrode. With the accumulation of charge at the two poles, the internal electric field strength increases continuously until the internal field strength exceeds the minimum field strength of ionization. The gas is broken down, and the conductivity increases rapidly, forming a voltage spike.

After obtaining the accurate *R*(*T*) curve, we can draw the *R*–*q* curve, as shown in Figure 8. It can be seen that under different voltages, the energy absorption curves of the two samples differ minimally over a larger range. There is an obvious turning point at *R*_2_ = 0.172 Ω in d*R*/d*q*. Because the resistance at this turning point is far greater than R_1_ at the end of liquefaction mentioned above, the turning point must be caused by the liquid–gas phase transition process. Before foil gasification, the volume expansion of the foil is very small. In the case of high current heating, the heat conduction of the foil does not need to be considered. Therefore, the absorbed heat energy on the foil is almost all used to heat the metal:(4)$$dq=mC_{p}\left(T\right)dT=I^{2}Rdt$$
where *m* is the mass of the foil, *t* is the temperature of the foil, and *I* and *R* are the current and the resistance of the foil, respectively. As the mutation points of d*R*/d*q* of the two samples in the above figure are very close, the resistivity and gasification temperature of the samples at the beginning of gasification are similar. Under the same voltage, the energy and energy difference absorbed by the two samples at turning points *R*_2_ and *R*_max_ are shown in Table 3.

It can be seen that with increasing voltage, the absorption energy of the samples at *R*_2_ changes little, while the final absorption energy of the samples at *R*_max_ increases with increasing voltage. The difference between the solid properties of the two samples disappears completely after sample gasification, so when the resistance reaches the *R*_2_ point, the energy absorbed by the same sample (*E_R_*_2_) mainly depends on the functional relationship of *C*_p_(*T*). However, the volume change of the same sample in the solid–liquid phase can be ignored. *C*_p_(*T*) hardly changes with changing heating rate, so the change in *E_R_*_2_ is very small.

After the foil resistance reaches R_2_ and starts to gasify, its volume expands rapidly, accompanied by dramatic changes in pressure and density. The higher the initial voltage is, the faster the energy input is. The energy input rate determines the conversion rate of the liquid phase to the gas phase. Because gas expansion is a slower process than boiling phase transformation, the faster the energy input is, the greater the peak pressure that the sample can reach in the process of electric explosion, the faster the subsequent expansion process, and the higher the temperature increase rate of the gas. The differences in pressure and temperature in the process of gas expansion lead to differences in the *q*-*R* curves of the same type of sample. It can be seen from the curve in the figure that for the same type of sample, the higher the initial voltage is, the smaller the peak resistance and the smaller the d*R*/d*q*.

In summary, when the resistance reaches the maximum value, the difference between the final absorption energies (Δ*E_R_*_max_) of the two samples is mainly due to two reasons: (1) The energy required for heating and destroying the lattice is different because of the differences in the microstructures, and the energy consumed for the MS sample is lower than that for the EP sample; (2) The difference between the energy absorption capacities is caused by the difference between the melting and gasification areas in the samples: the MS sample participates in the melting process over a wider area, and the high resistance area expands, so there is a greater peak resistance, but the high resistance in the melting area reduces the current density in the bridge area during gasification, so the area ultimately involved in the gasification process is smaller than that of the EP sample, and the MS sample absorbs more energy in the gasification stage. Because of the decrease in the current density in the gasification center, the ionization temperature (*T*_MS_) of the foil gasification center of the MS sample is lower than that (*T*_EP_) of the EP sample. Because the difference between the solid properties of the two samples disappears after gasification and the initial voltage is the same, the gas expansion processes are almost identical. Since the difference between the time of gas expansion and the initial energy before ionization is very small, it can be considered that the pressure of the two gases at the time of ionization is the same. Thus, we have:(5)$$\frac{E_{\mathrm{EP}}}{E_{\mathrm{SM}}}=\frac{\delta_{\mathrm{MS}}}{\delta_{\mathrm{EP}}}=\frac{T_{\mathrm{SM}}}{T_{\mathrm{EP}}}<1$$

Therefore, the average free path of the MS sample is smaller than that of the EP sample, so the minimum field strength required for ionization of the EP sample is smaller than that required for ionization of the MS sample. However, the final ionization voltage depends on the minimum distance d between the non-gasified regions. In this paper, although the d of the EP sample is larger, the field intensity needed for ionization is small enough, and the final ionization voltage is less than that of the MS sample.

To explain the difference between the results for the above two kinds of samples, the following single-layer model is established (as shown in Figure 9) according to the microstructure of the samples. Because the electric field intensity on the microstructure is uniform, the time scale of the electric heating process is far smaller than that of heat conduction, so the interlayer current and heat conduction can be ignored. The two samples can be regarded as three parts: the series structure of the Cu layer, Ni layer and premixed layer. Because the premixed layer is composed of micron-scale crystals of Cu and Ni, the premixed layer can be regarded as the series connection of Cu and Ni blocks in the same ratio of 2:3. Therefore, the resistance of the multilayer film in all eight periods on the plane micro-element (square) can be expressed as:(6)$$R=\frac{1}{8}/\left(\frac{h_{1}}{\rho_{1}}+\frac{h_{2}}{0.4\rho_{2}+0.6\rho_{3}}+\frac{h_{4}}{\rho_{4}}\right)$$
where *h_i_* is the thickness of each area and *ρ_i_* is the resistivity of each area. Through a literature review, the constant pressure heat capacities *C*_p_ of Cu and Ni and the functions of resistivity *ρ* with temperature *T* were obtained, where resistivity is a linear function of temperature. At 293 K, the resistivity of Cu at atmospheric pressure is 1.678 × 10^−8^ Ω·m, and the resistivity temperature coefficient is α = 0.0045. The resistivity of nickel is 6.84 × 10^−8^ Ω·m, and the temperature coefficient of resistivity is α = 0.0069. Therefore, the resistivity of Ni is larger than that of copper, and the resistivity increases faster with increasing temperature. Because the resistance of the foil is very small in the solid phase, for the RLC circuit, the current growth rate in the initial phase is proportional to the time, and its slope is *U*_0_/*L*_0_. According to the cross-sectional area of the foil, the current density I on the foil can be obtained as a function of time. Through the parallel series circuit to solve the energy distribution, combined with the temperature increase equation of the foil electric heating, the curve of the foil resistance growth with time as shown in Figure 10 and the curve of the temperature growth with time in four regions of the model as shown in Figure 11 are obtained under different voltages and thicknesses of the premixed region (it is calculated until one of the four regions reaches its melting point).

We use the ratio of the premixed layer obtained from the previous analysis to simulate the electrical heating process of the micro element. a = 0.17 for EP samples and a = 0.34 for MS samples, respectively. The growth rates of resistance are from the 4000 V-MS sample, 4000 V-EP sample, 2500 V-MS sample and 2500 V-EP sample, which is consistent with the growth trend of the actual total resistance curve in Figure 4. This is because the resistivity of the two metals is different. Under the multilayer parallel structure, the electrical energy is more likely distributed in the Cu layer with low resistivity. The existence of the premixed region helps heating the high-resistance Ni material in the premixed region. Due to the higher temperature coefficient of the resistance of Ni, the overall resistivity increases very fast due to the increase in the resistivity of Ni, and the efficiency of the energy conversion is also greatly improved.

It can be seen from the temperature increase curve in Figure 10 that the priority of electric energy allocation in the four parts of the model is ① ④ ③ ②, among which Region ① melts first. Since the resistivity and resistivity temperature coefficient of Ni are larger than those of Cu, the electric energy distributed by Cu is more in parallel structure, and that of Ni is more in series structure. Because the current growth rate is almost constant in the solid phase, the increase in the sample resistance improves the overall energy conversion efficiency of the sample. Therefore, we can also draw a conclusion similar to Figure 10: the emergence of the premixed layer redistributes the electric energy. As the premixed layer is thinner in EP sample than that in MS sample, the current easily flows through the Cu with low resistivity, making the overall energy conversion efficiency of the EP sample microstructure lower, and the heating rate of the four parts of the EP sample is lower. When the premixed layer is thick, the current distributes more energy to Ni through the series part of the premixed layer, and due to the thinning of Cu and Ni layers, the energy conversion efficiency of each layer is improved, so the heating rate of each layer is increased. Therefore, the energy conversion efficiency of the MS sample with a thick premixed layer in the solid phase is significantly better than that of the EP sample, which accelerates the process of electric explosion and advances the peak time of resistance.

## 4. Conclusions

In this paper, the structure and electrical explosion performance of multilayer films were studied and compared under the same thickness, modulation period and size. Among them, the crystal particle size of the Cu/Ni multilayer film prepared by the magnetron sputtering method is large, the premixed layer between layers is thick. The crystal particle size of the Cu/Ni multilayer film prepared by the electroplating method is small, the boundary between layers is clear, and the thickness of the premixed layer is thin. Based on the analysis of the current and voltage data, the curves of the resistance versus time and the resistance energy curves of different samples under different voltages are obtained. Combined with the experimental images after the foil electric explosion, we find that the difference in the electric explosion performance is mainly due to the difference between the absorption energy efficiencies of the solid phases as a result of the different structures of the two samples. The energy conversion efficiency of the MS sample in the solid phase is better than that of the EP sample, which accelerates the whole process of the electric explosion and advances the resistance spike. We also find that the MS samples absorb more energy in the solid and liquid phase and the high resistance area before gasification is larger. As the melting process of the MS sample occurs over a larger area, the size of the gasification area is reduced, and the current density of the gasification area is lower than that of the EP samples. Therefore, the temperature of the MS sample is lower, and the average free path of electrons is smaller when this sample is ionized. Therefore, the minimum field strength required by the MS sample is higher than that required by the EP sample.

To explain the difference in the electric exploding performance of the two kinds of samples, a single-layer micro-model of the multilayer foil was established. Through the simulation of the heating process of each part of the model, we find that the current flowing through the high resistance material can be increased by the model premixed layer. It is shown that the high energy conversion efficiency of the solid phase of the MS sample is due to the thickness of the premixed layer between the layers being thicker. Thus, the increase in the thickness of the premixed layer can greatly improve the energy conversion efficiency of the solid phase of the sample. In this study, the relationship between the microstructure of multilayer film exploding foils and their electrical explosion performance is discussed from the two aspects of experiment and theory, which is helpful for regulating the electrical explosion performance by controlling the microstructures of multilayer film exploding foils in the future.

## Figures and Tables

**Figure 1 micromachines-11-00528-f001:**
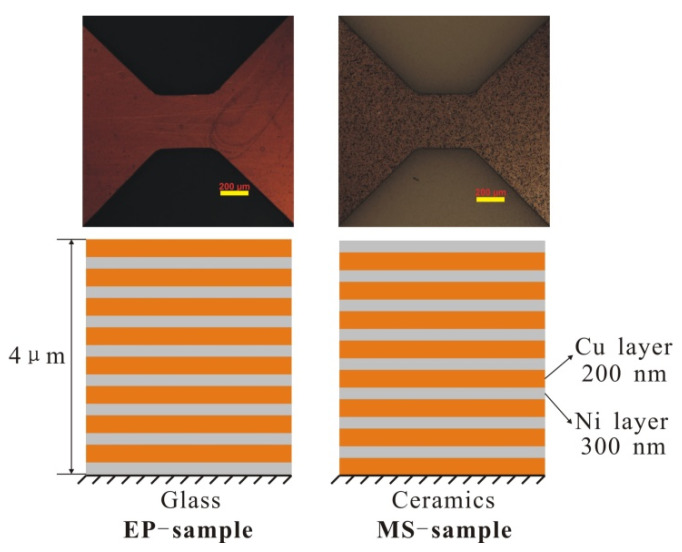
The initial morphology (face view, scale bar: 200 nm) and the diagrammatic sketch (side view) of two samples.

**Figure 2 micromachines-11-00528-f002:**
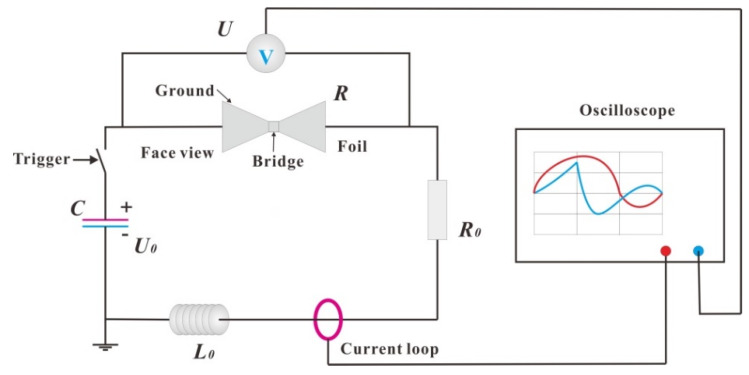
Electric explosion experimental system for multilayer foil. (*R*—foil resistance; *U*—foil voltage; *R*_0_—line resistance; *U*_0_—initial charging voltage; *L*_0_—loop resistance.).

**Figure 3 micromachines-11-00528-f003:**
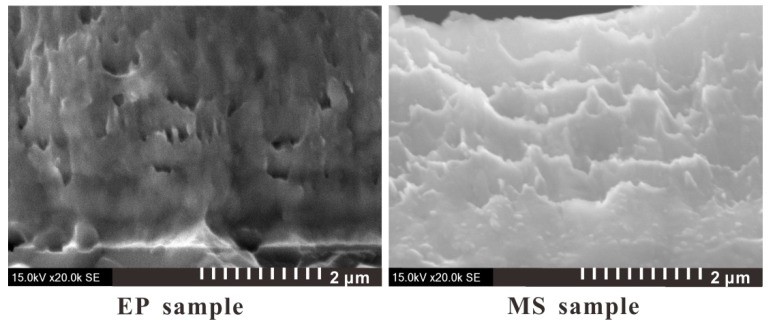
SEM diagram of the foil section of the multilayer bridge (side view, scale bar: 2 μm).

**Figure 4 micromachines-11-00528-f004:**
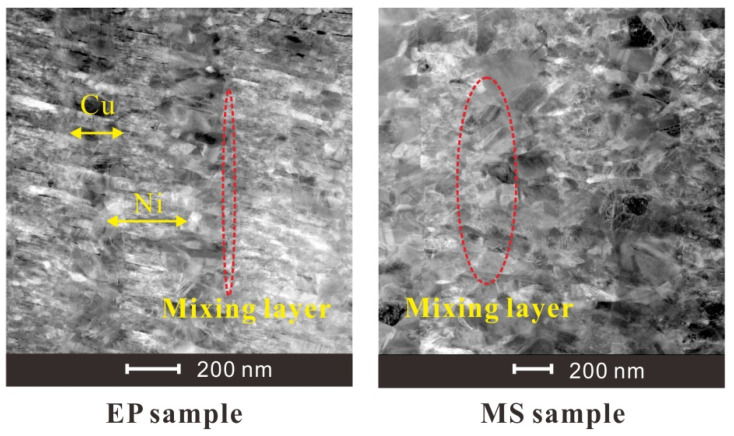
TEM diagram of the foil section of the multilayer bridge (side view, scale bar: 200 nm).

**Figure 5 micromachines-11-00528-f005:**
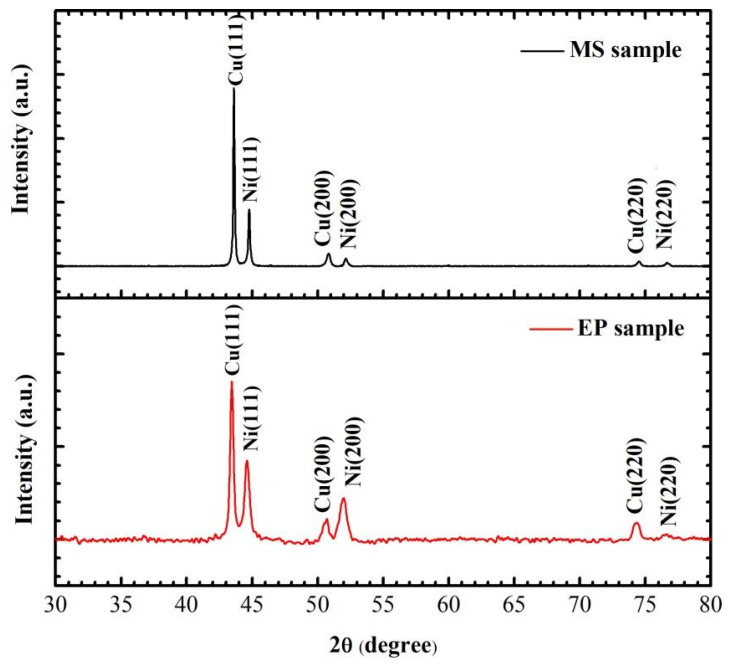
Comparison of XRD peaks of EP and MS samples.

**Figure 6 micromachines-11-00528-f006:**
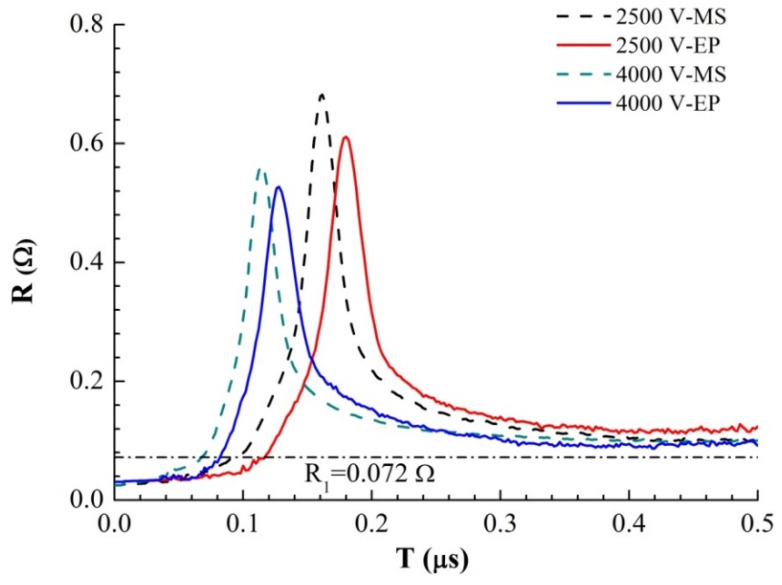
Resistance–time curves of the two samples under different *U*_0_.

**Figure 7 micromachines-11-00528-f007:**
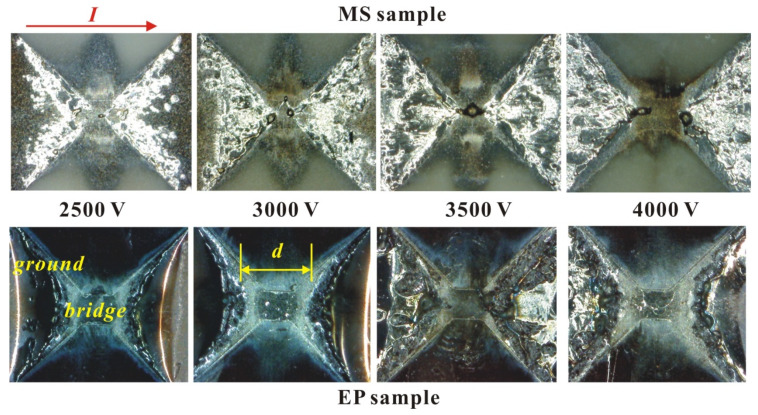
Morphological characteristics of two samples after electric explosion.

**Figure 8 micromachines-11-00528-f008:**
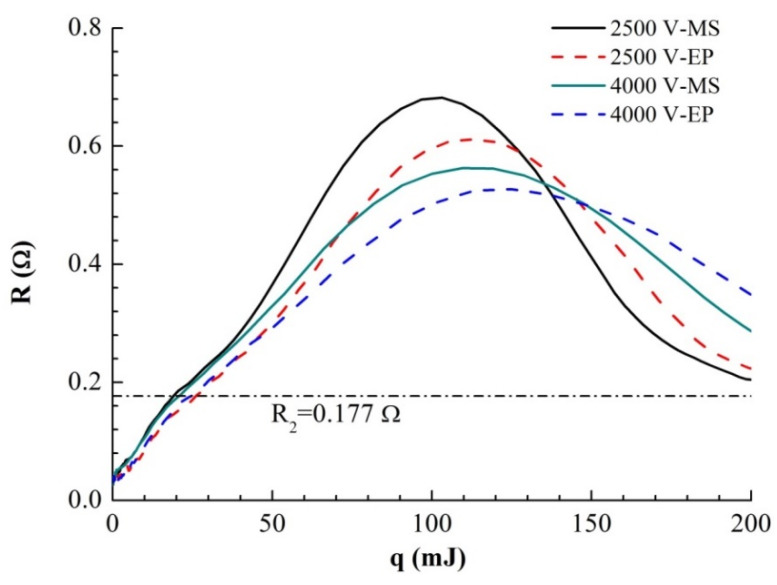
R (resistance)–q (absorption energy) curves of two samples under different *U*_0_.

**Figure 9 micromachines-11-00528-f009:**
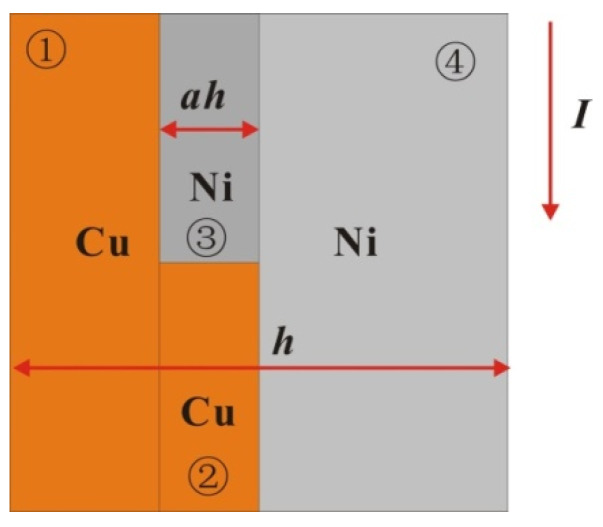
Single-layer mixing model of multilayer foil (horizontal direction is its thickness direction).

**Figure 10 micromachines-11-00528-f010:**
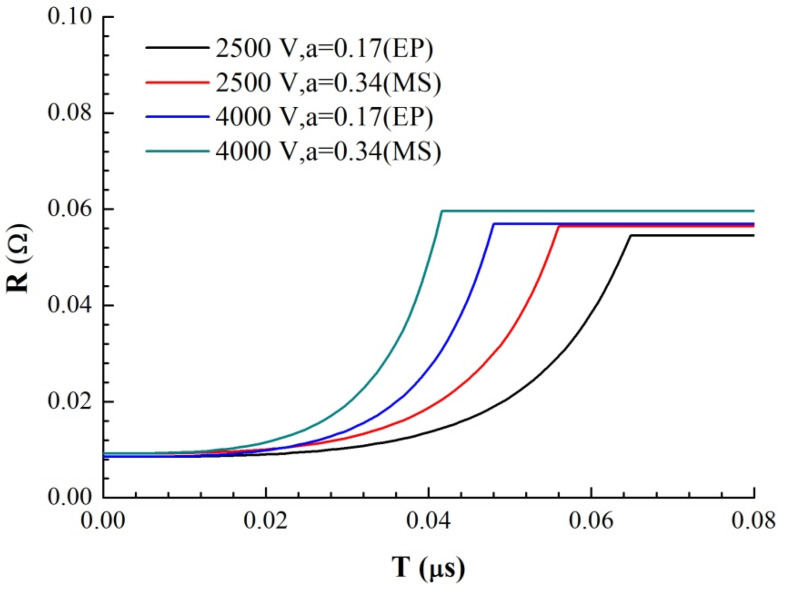
Theoretical simulation of the resistance curves of two samples with time under different *U*_0_.

**Figure 11 micromachines-11-00528-f011:**
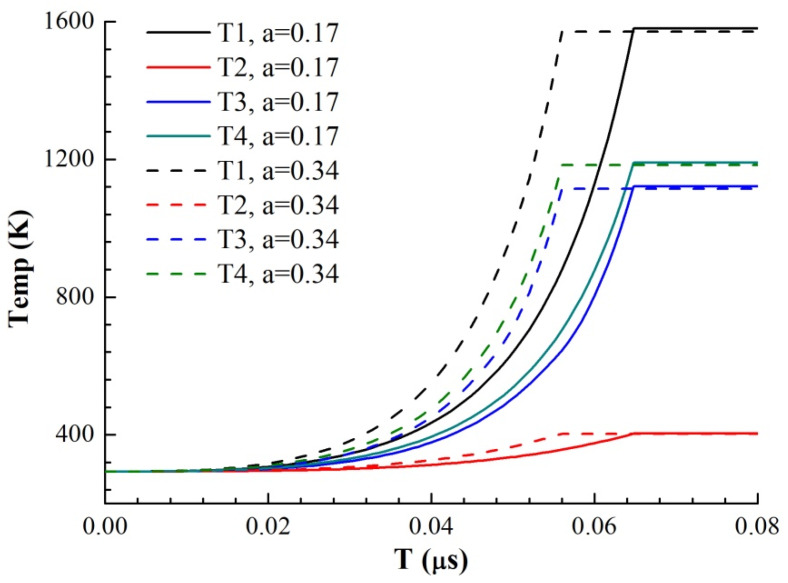
Simulation results of the temperature–time curve of each part of the two samples with different mixing layer thicknesses (*U*_0_ = 2500 V).

**Table 1 micromachines-11-00528-t001:** Grain size calculated from XRD data.

	2θ (°)	FWHM-EP (β)	D (nm)	FWHM-MS (β)	D-MS (nm)
Cu(111)	43.23	0.0053	27.99	0.0025	59.95
Ni(111)	44.59	0.0060	19.56	0.0035	42.67

**Table 2 micromachines-11-00528-t002:** Peak voltage of two samples under different *U*_0_ at *R*_max_.

*U*_0_ (V)	MS-*U_max_* (V)	EP-*U_max_* (V)
2500	1659	1569
3000	1702	1653
3500	1888	1656
4000	1844	1771

**Table 3 micromachines-11-00528-t003:** Energy differences between the two samples at *R*_2_ and *R*_max_ (the unit of energy is mJ).

*U*_0_ (V)	*E_R_*_2_-MS	*E_R_*_2_-EP	Δ*E_R_*_2_	*E_R_*_max_-MS	*E_R_*_max_-EP	Δ*E_R_*_max_
2500	19.08	26.08	7.00	100.78	111.96	11.18
3000	20.03	22.38	2.38	104.80	116.59	11.79
3500	20.01	24.42	4.41	114.54	115.68	1.14
4000	20.09	24.48	4.38	113.24	122.99	8.75

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
