# Peer review of "Study on Electrical Explosion Properties of Cu/Ni Multilayer Exploding Foil Prepared by Magnetron Sputtering and Electroplating"

_micromachines, 2020, doi:10.3390/mi11050528_

Round 1
Reviewer 1 Report
The article proposes the fabrication of a multilayer film of Cu/Ni prepared by two different methods focusing on the study on electrical explosion properties, but in its present form is not very clear for the reader, there are very simple errors that reveals the manuscript has not been controlled before submitting. More in general, I have a lot of doubts related to the results reported in the article as better explained point by point in the following.
- Introduction is not focused on the problem. Why is it important to study the electrical explosion properties? Authors should better introduce the problem and better explain their innovation. In general, I would like to read a more focused introduction, where the last paragraph points out the novelty of the work proposed.
- Materials and methods: some more details are needed
Results and discussion:
- Figure 2 is not very simple to read as authors state in the text. They could try to improve the description. Scale bar are too small.
In general authors don’t furnish values in numbers, like the dimension of the mixing layer and often use adjective (i.e. small ) try to be more precise.
- Figure 3 the mixing zone is not really visible, what about the crystalline structure of Ni and Cu under the EP and MS process?
- Form Figure 4 to the end of the article, the authors state all their comments on a very small difference in terms of results between the two distinct methods of fabrication. For example the difference of the plot of R/ time shows difference of order of less than tens nanosecond. What about the errors of all the measure reported in text? So small difference could be covered by errors of measurements, no reference to that could be found in the text. In my opinion this is a very important data to points out.
- Figure 5: please introduce a reference to time zero.
- page 5 line 153, symbols are confused
- Page 6 line 185, the reference to the figure is not correct, there are a lot of errors of this type along the manuscript.
- The results of the theoretical simulation are too much report very similar curves for the different conditions, too much similar, please add more information on this point.
In general I find a lot of errors of in text and also the title is repetitive in its present form.
Reviewer 2 Report
Lei et al. reported the study on the electrical explosion properties of Cu/Ni multilayer. The authors present an in-depth study and I recommend the acceptance of this manuscript for publication after minor revision. Few points have to make it clear.
(1) If possible, the author can provide the EDM mapping of the Cu/Ni layers, which will make the growth of the layers very much clear.
(2) Single-layer mixing model of the multilayer foil (Figure 7), if not looks towards the real picture. The authors must make it more realistic.
Reviewer 3 Report
The present study synthesized Cu/Ni laminates films via two different methods of electroplating and magnetron sputtering, and investigated the electrical explosion properties of them. The manuscript was well-prepared and could be accepted as a paper in Micromanchines. A small question: the quality of SEM images shown in Figure 2 seems to be low. An additional schematic cartoon should be helpful to demonstrate the laminated structure. The same issue also exists in Figure 3.
Round 2
Reviewer 1 Report
The manuscript in the current form is improved, especially referring to the experimental section.
Some points still need to be fixed:
-1 My first point was not answered by the authors "Introduction is not focused on the problem. Why is it important to study the electrical explosion properties? ". I think that adding this point could improve the introduction for a general reader.
-2 Please review all the captions adding some more information useful for the comprehension of the Figure. Some scale bar are still not readable.
-3 Some typos and repetitions are still present in the text.
